# Heat Transfer Investigation during Condensation on the Horizontal Pipe

**Elza R. Zainullina * and Vladimir Yu. Mityakov**

Science Educational Center "Energy Thermophysics", Peter the Great St.Petersburg Polytechnic University (SPbPU), St. Petersburg 195251, Russia
* Correspondence: zajnullina_er@spbstu.ru; Tel.: +7-812-552-7773

**Abstract:** This paper presents an experimental investigation of condensation heat transfer by gradient heatmetry. The experiments were carried out during the condensation of saturated steam at atmospheric pressure on the cooled surface of a horizontal pipe. The distributions of the local heat flux, surface temperature, and heat transfer coefficient along the circumference of the horizontal pipe were experimentally determined. The surface average condensation heat flux on the horizontal pipe was about $141.06 \text{ kW/m}^2$. The proposed method allows us to determine the area of condensate accumulation on the pipe (in the range of azimuth angle $\varphi = 150\ldots180°$) in which the heat flux decreases by 34% of the average value. The heat flux per unit area relative uncertainty was about 5.2%. The surface-averaged heat transfer coefficient during condensation on the horizontal pipe was about $5.5 \text{ kW/(m}^2 \times \text{K)}$, and relative uncertainty was about 9.4%.

**Keywords:** condensation heat transfer; gradient heatmetry; single horizontal pipe; heat transfer coefficient distribution; heat flux fluctuations

## 1. Introduction

Recuperative heat exchangers are important parts of the heat-removing systems which are used at thermal and nuclear power plants. Important tasks such as improving the heat exchanger effectiveness or determining optimal operating regimes can be solved by studying heat transfer during condensation.

The purpose of this work is to apply an innovative approach—gradient heatmetry [1]—to the study of heat transfer during condensation. Currently, various experimental approaches are used to determine the heat flux and heat transfer coefficients (HTC) during condensation on geometrically enhanced surfaces [2–5], during condensation of a vapor–air mixture [6–14], or condensation on a surface with a hydrophobic coating [15,16], etc. Despite the variety of tasks and objectives of research, we single out the most common experimental approaches and the accuracy of estimating the heat flux and HTC.

Most experimental studies of heat transfer during condensation are carried out using temperature measurements [6–14]. The advantages of this approach are the design simplicity and availability of sensing devices. However, the method is invasive and requires special installation of sensors to reduce distortion of the natural condensate flow. In addition, to reduce the error in temperature measurement, the thermocouple wires should be retracted along the isothermal surface to a length of 150...200 thermocouple junction diameters [17].

Heat transfer during condensation is most often determined by the following approaches:

- The heat flux is assumed to equal the rate of heat absorption by the coolant

$$q = \frac{\dot{G}_w (h_{out} - h_{in})}{F},$$ (1)

where $\dot{G}_w$ is the coolant water mass flow rate, $h_{out}$ and $h_{in}$ are the coolant enthalpy rise at the temperature of the coolant at the inlet and outlet of the pipe, and $F$ is the area of the heat exchange surface.

- The heat flux is calculated through direct measurement of the condensate mass flow rate from the heat exchange surface

$$q = \frac{\dot{G}_c i_{fg}}{F}, \tag{2}$$

where $\dot{G}_c$ is the condensate mass flow rate, and $i_{fg}$ is the latent heat of condensation.

- The heat flux is assumed to equal the longitudinal conductive heat flux in the condensing wall

$$q = k\frac{\Delta T_w}{\Delta x}, \tag{3}$$

where $k$ is the thermal conductivity of the wall, and $\frac{\Delta T_w}{\Delta x}$ is the temperature gradient.

- The heat flux is calculated according to Newton's cooling law

$$q = \frac{T_w - T_{cw}}{R}, \tag{4}$$

where $T_w$ is the wall temperature, $T_{cw}$ is the cooling water temperature, and $R$ is the overall thermal resistance.

The heat transfer coefficient (HTC) averaged over the pipe surface can be expressed as:

$$h = \frac{q}{T_s - T_w}, \tag{5}$$

where $T_s$ and $T_w$ are the temperature of the steam and pipe surface.

In the investigation of G. Fan et al. [6], heat flux was calculated by Formula (1). The investigation was devoted to developing a new dependence that can reliably reflect the complex dependencies of the condensation HTC on pressure, air mass fraction, and wall supercooling. The dependence is based on the experimental results carried out using thermocouples installed at nine axial points along the test section. A total of 18 K-type thermocouples were installed at the surface of the pipe. The authors indicated that the relative uncertainty of the surface-averaged HTC was about 7%.

In the work of Y.-G. Lee et al. [7], the local heat transfer during condensation of a vapor–air mixture on the vertical pipe surface was calculated by Formula (3). For this, 12 K-type thermocouples were installed in 6 sections along the length of the vertical pipe to measure the temperature on the outer and inner pipe' surfaces. The authors intentionally increased the wall thickness of the pipe to 5 mm compared to other experiments to reduce errors. Thermocouples were silver soldered into drilled holes, one about 4.5 mm deep for the inner wall and the other about 1.0 mm deep for the outer wall. Unfortunately, the authors did not provide information about the extraction of thermocouple wires. The relative uncertainty of the local condensation HTCs was estimated to be 13.6%.

Swartz M.M. and Yao Sh.-Ch. [10] implemented three independent methods with Formulas (1)–(3) for determining the heat flux during film condensation on the plate surface. According to the authors, measurement using the condensate mass flow rate is the most accurate method for determining the HTC. The relative uncertainty of the condensation HTCs was estimated to be about 5%. The authors indicated that an increase in relative uncertainty up to 9% is possible with a decrease in the temperature difference between the steam and wall.

In the investigation of steam-air condensation on containment vessel made by Chen R. et al. [13], the heat flux was determined with Formula (3). The measuring system mainly consisted of 12 thermocouple probes which consisted of 3 T-type thermocouples: 1 was set in the coolant channel and the other 2 were located in different depths of the test

plate. When the thermal equilibrium was achieved, the condensation heat flux equaled the longitudinal conductive heat flux in the condensing wall. The maximum relative uncertainties of the average heat flux was 6.1%, and HTC was about 10.4% within the experimental parameter ranges.

The literature review showed that the most common method for determining the heat flux is the calculation through the mass flow rate and the change in the enthalpy of the cooling water [3,18–25]. The main advantages of this approach are as follows:

1.  The method allows us to determine the heat flux during condensation on various surfaces.
2.  It allows us to measure the flow of condensate without the need to separate the main one, formed on the experimental model, and the additional one, formed on the setup case.
3.  The method leads to a decrease in the number of measuring probes.

The irremediable defect of this method is the possibility of determining only the surface-averaged heat flux and HTC.

The temperature measurement application for the study of heat transfer during condensation on a horizontal pipe becomes more difficult since the condensate film thickness changes along the circumference of the pipe. Zhang J.X. and Wang L. [14] used thermistors devoted to the accumulation of air during the condensation of vapors across a horizontal pipe. In two sections of a horizontal pipe, eight sheath PT100 thermistors were welded into a T-shaped blind. Among the results of the study was the HTC distribution around the circumference of the pipe during condensation. The local condensation heat flux across the horizontal tube was obtained according to Formula (4). The standard error of the local HTC on the condensation side was estimated by the authors at 2.5–10%.

An alternative approach to studying heat transfer during condensation is the use of optical methods [10,26,27]. For example, Ölçeroğlu, E. et al. [26] and Stevens K.A. [27] used an optical microscope to determine the heat flux during droplet condensation. The authors measured the volume of the resulting condensate as a measure of latent energy transfer. The heat transfer rate was calculated by linear approximation of latent energy as a function of time, and the average heat flux was calculated by dividing of the area of the field of view, resulting in a heat flux of 53...79 $W/m^2$. This approach makes it possible to obtain a heat flux with an uncertainty of about 2 $W/m^2$.

There are significantly fewer papers on the study of heat flux during condensation using heat flux sensors [28–30]. In the investigation of laminar convective condensation of pure vapor inside an inclined circular tube by Lyulin Yu. et al. [28], the heat flux was determined by three independent methods by Formulas (1) and (2) and on the direct heat flux measurements using the heat flux sensors. The authors used the heat flux sensors produced by Omega Company with sizes: width 28.5 mm, length 35.1 mm, thickness 0.178 mm. Considering that the inner diameter of the pipe was 4.8 mm (the outer diameter of the pipe was not specified), the use of heat flux sensors of such dimensions seems inappropriate. The authors did not indicate the uncertainty of measuring the heat flux using the heat flux sensor. The relative uncertainty of the HTC measurements was estimated as ±8%.

In the study of Janasz F. et al. [29] the authors presented developments in the field of measuring heat flux during reflux condensation using heterogeneous gradient heat flux sensors (HGHFS) made of stainless steel and nickel. The condenser tube made of stainless steel had an inner diameter of 20 mm and a wall thickness of 5 mm. The HGHFS with a cross area of 100 $mm^2$ was located very close (0.5 mm) to the inner surface of the wall. This way of the sensor installation did not allow studying the heat flux fluctuations during condensation and reduced the information content. The use of an HGHFS made of a steel–nickel composition with a relatively low sensitivity of 0.4 mV/W required the use of a sensor with a large area. This approach made it possible to reduce the relative uncertainty of heat flux measurement to 2.86%.

We assume that the use of heat flux sensors to measure the heat flux during condensation is not common due to the large area of the sensing element which is installed on the model surface.

This investigation proposes a fundamentally new approach to determining the heat flux during condensation using gradient heatmetry. This approach makes it possible to measure the heat flux directly according to the records of the gradient heat flux sensor (GHFS). The GHFSs used in gradient heatmetry are practically inertialess for the study of condensation heat transfer because their response time is about $10^{-8} \ldots 10^{-9}$ s. The type of GHFS, its size, and volt–watt sensitivity can be selected based on the goal and objectives of the study. The investigation peculiarity lies in the selection of the optimal type of GHFS, determining the required dimensions of the sensor, and ensuring the method of installation on the pipe.

## 2. Experimental Procedure

### 2.1. Gradient Heatmetry

Gradient heatmetry [1] was developed at the Science Educational Center "Energy Thermophysics", Peter the Great St.Petersburg Polytechnic University for direct measurement of the local heat flux. This method is based on the use of GHFS which is made of materials with anisotropy of thermophysical properties. The GHFSs operating principle is based on thermo-EMF generation under the action of a heat flux on an anisotropic material. The value of thermoEMF (signal generated by the GHFS) is proportional to the heat flux passing through the sensor:

$$E = qS_0A \tag{6}$$

where $q$, W/m$^2$ is the heat flux; $A$, m$^2$ is the GHFS's cross area, and $S_0$, mV/W is the GHFS's volt–watt sensitivity.

The scientific group has developed two types of GHFSs. The first type (Figure 1a) includes GHFSs created from single-crystal bismuth which has a natural anisotropy of thermophysical properties. Another type is based on the artificial (synthetic) anisotropy of properties which occurs when dissimilar metals are combined in one composition. These sensor types are called heterogeneous GHFS (HGHFS) (Figure 1b).

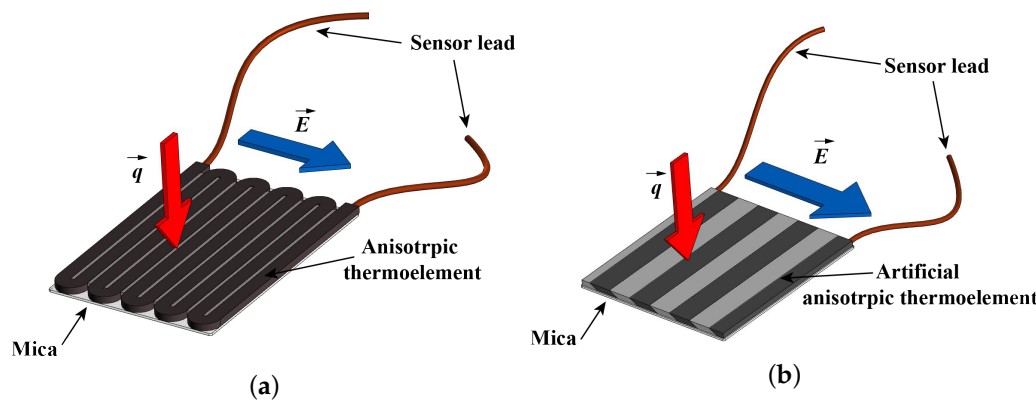

**Figure 1.** Scheme of GHFS (**a**) and HGHFS (**b**).

Each sensor type has its own advantages and disadvantages. For example, the volt–watt sensitivity of GHFSs does not depend on the surface temperature, but their application is limited by the melting point of bismuth (about 544 K). The GHFSs are successfully used in the study of forced convection [31], but they cannot be used for high-temperature experiments. At the same time, the HGHFSs with copper–nickel composition have a higher temperature limit (about 1300 K), but their volt–watt sensitivity is lower than GHFSs and depends on the temperature. HGHFSs are used in high-temperature experiments such as boiling on a superheated surfaces [32].

Both sensor types are applicable to the steam condensation heat transfer study because the experiments were carried out at atmospheric pressure, and the steam temperature was about 100 °C. Therefore, the first part of the experiments was devoted to sensor-type selection.

### 2.2. GHFS Type Selection

The GHFS type selection was carried out on a special experimental setup (Figure 2). The setup consisted of two coaxial pipes: the inner one was made of copper ($d$ = 18 mm outer pipe diameter, $\delta$ = 1 mm wall thickness), and the outer one (the case) was made of heat-resistant plastic. The inner tube was fixed in the case by the rubber flanges which expanded upon heating, additionally pressurizing the measuring section. The saturated steam was supplied to the annular space, and cooling water with a temperature of 293 K was supplied inside the copper pipe. The experimental setup was located horizontally.

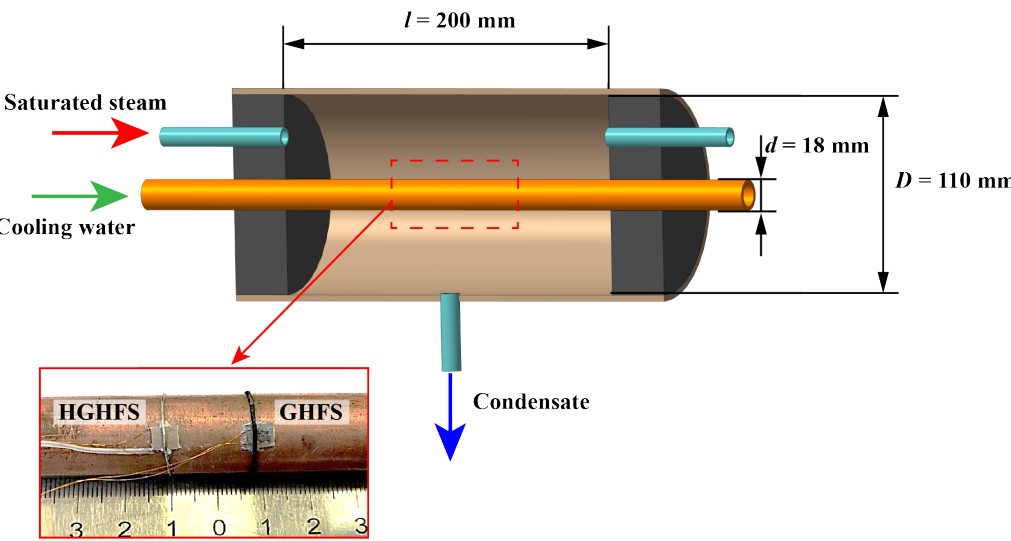

**Figure 2.** The experimental setup for GHFS-type selection.

The GHFS, HGHFS made of copper–nickel and the L-type thermocouple were installed on the upper generatrix of the pipe. Characteristics of the heat flux sensors are presented in Table 1.

**Table 1.** Sensor specifications.

| Sensor Type | Material | Sizes, mm | Volt-Watt Sensitivity $S_0$, μV/W Depending on Temperature | |
|---|---|---|---|---|
| | | | $T$ = 322 K | $T$ = 372 K |
| GHFS | single-crystal bismuth | $3 \times 3 \times 0.2$ | 2810 | 2810 |
| HGHFS | copper–nickel composition | $5 \times 5 \times 0.2$ | 21.5 | 19.3 |

The following measurements were made to select the sensor type:

1.  The local heat flux by the GHFS ($q_{GHFS}$) and HGHFS ($q_{HGHFS}$).
2.  The temperature of the copper pipe outer surface ($T_{surf}$) by the L-type thermocouple.
3.  The temperature difference ($\Delta T$) between inlet and outlet cooling water by the L-type differential thermocouple.
4.  The cooling water ($\dot{G}_w$) and condensate ($\dot{G}_c$) rate.

The time-averaged experimental results are presented in Table 2.

**Table 2.** The time-averaged results of experiments on selecting the sensor type.

| No | $q_{GHFS}$ | $q_{HGHFS}$ | $T_{surf}$ | $\Delta T$ | $\dot{G}_w$ | $\dot{G}_c$ |
|---|---|---|---|---|---|---|
| | kW/m$^2$ | | | K | | g/s |
| 1 | 66.4 | 64.8 | 322 | 1.5 | 120 | 0.34 |
| 2 | 55.9 | 56.6 | 337 | 3.6 | 41 | 0.28 |
| 3 | 34.4 | 35.7 | 354 | 9.0 | 9.6 | 0.17 |
| 4 | 24.8 | 23.7 | 362 | 9.5 | 9.0 | 0.13 |
| 5 | 2.9 | 0 | 372 | 0 | 0 | - |

The difference between the time-averaged heat flux measured by GHFS and HGHFS did not exceed 5%. The experimental results confirmed that both sensor types are applicable to the time-averaged condensation heat transfer study. The GHFS type selection should be carried out, not on averaged results, but on the time-dependent heat flux graph, i.e., the heat flux dependencies on time. Figures 3 and 4 show the time-dependent heat flux graphs built according to measurements of the HGHFS and GHFS, respectively.

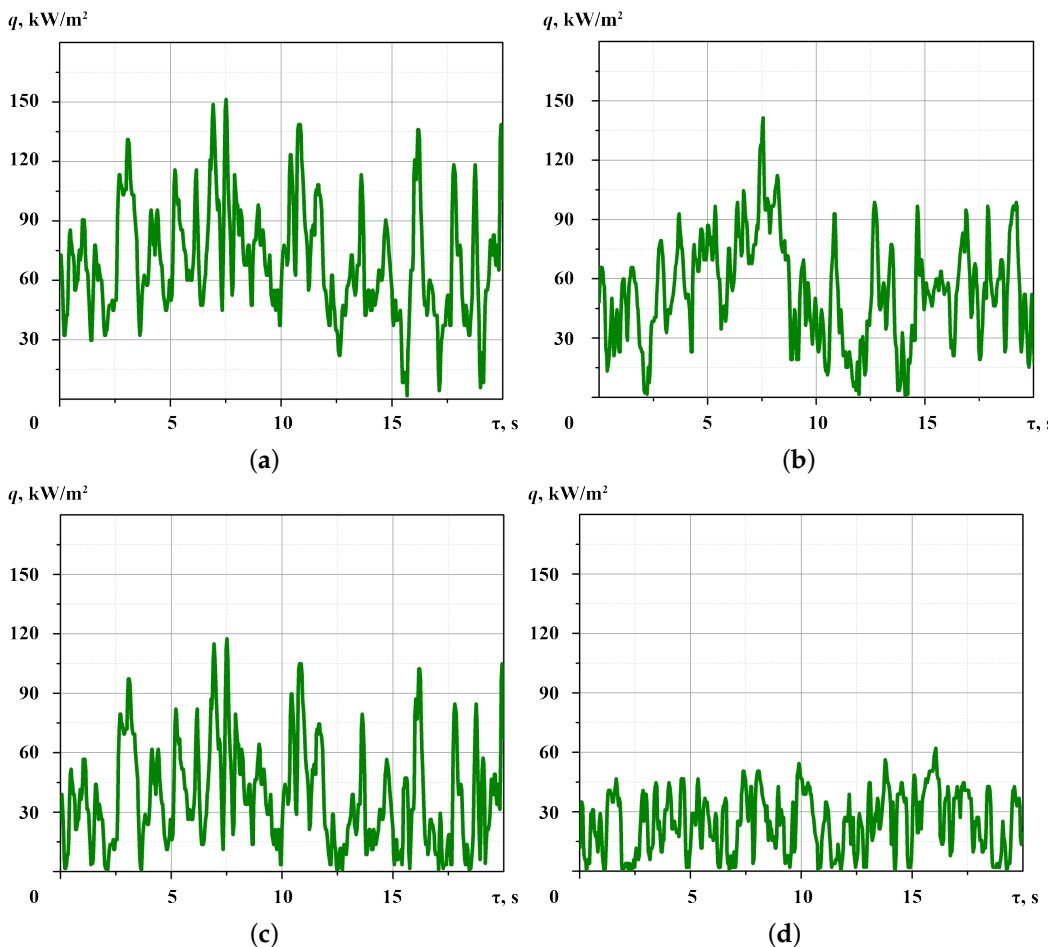

**Figure 3.** Time-dependent heat flux graph by HGHFS for condensation of saturated steam on the upper generatrix of the pipe at the surface temperature: (**a**)—322 K; (**b**)—337 K; (**c**)—354 K; (**d**)—362 K.

The time-dependent heat flux graph analyses (Figures 3 and 4) show the heat flux fluctuations registered by the HGHFS are higher than the same values measured by GHFS. For example, at a surface temperature of $T_{surf}$ = 322 K, the fluctuations amplitude according to the GHFS is up to about 40 kW/m$^2$, but fluctuations amplitude registered by the HGHFS is higher and reaches a value of about 135 kW/m$^2$. The time heat flux graph (Figure 3)

shows a decrease in the heat flux to zero which does not correspond to physical concepts of heat transfer during condensation. These effects are explained by the HGHFS's low volt–watt sensitivity. Studies have shown that HGHFSs can be used only to determine the time-averaged local heat flux.

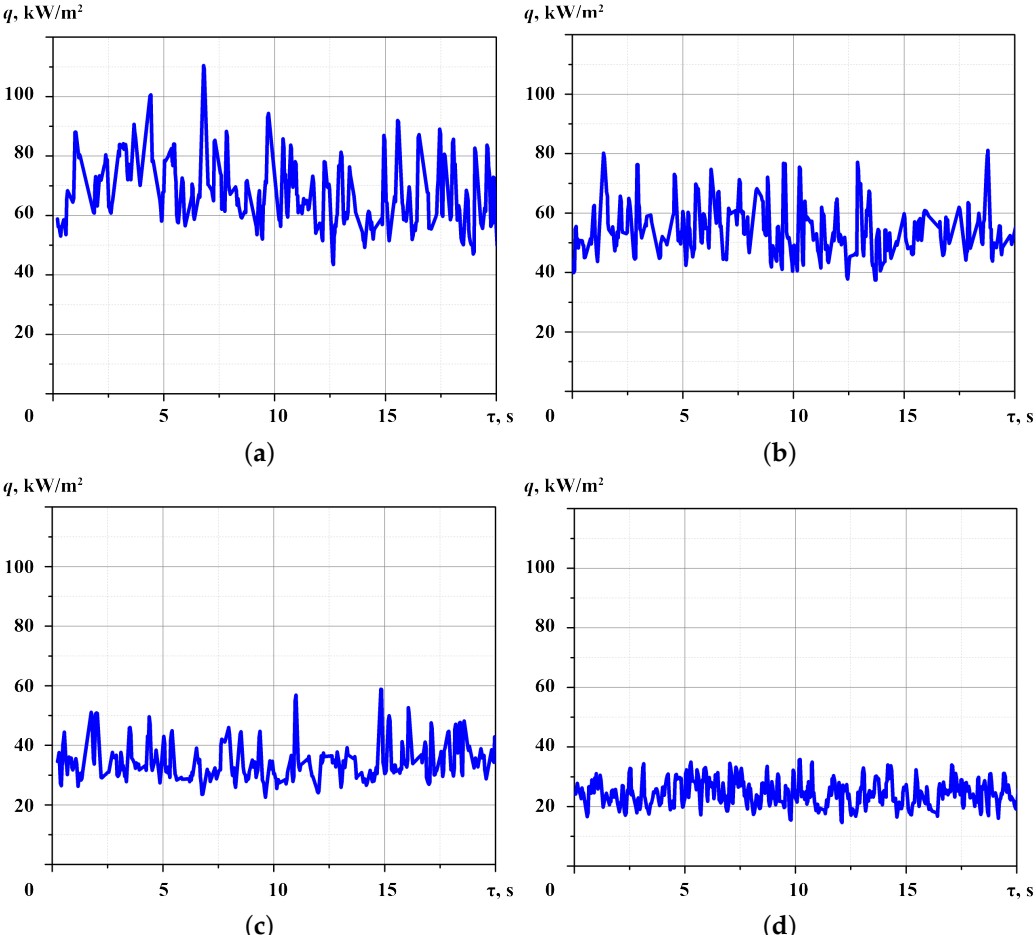

**Figure 4.** Time-dependent heat flux graph by GHFS for condensation of saturated steam on the upper generatrix of the pipe at the surface temperature: (**a**)—322 K; (**b**)—337 K; (**c**)—354 K; (**d**)—362 K.

Studies have confirmed that it is better to use GHFS with single-crystal bismuth in experiments on the condensation heat transfer study. According to the condensation heat transfer theory, the condensate film thickness on the pipe's upper generatrix is minimal, and there is no wave formation on its surface. So, the heat flux fluctuations are associated with the formation and coalescence of condensate drops and their rivulet run-off from the pipe's upper generatrix. This process occurs when the condensate rate is insufficient for the condensate film formation.

The GHFS high information content is shown on the heat flux graph corresponding to the regime without the measuring section cooling (Figure 5). In this case, the pipe's outer surface temperature is about 372 K, and the GHFS registers an average heat flux of 2.9 kW/m$^2$. The difference between steam and the outer pipe surface temperature is close to the temperature measurement uncertainty, so the temperature measurement is not informative.

It is recommended to use GHFS to measure the heat flux during steam condensation on the outer surface of a horizontal pipe.

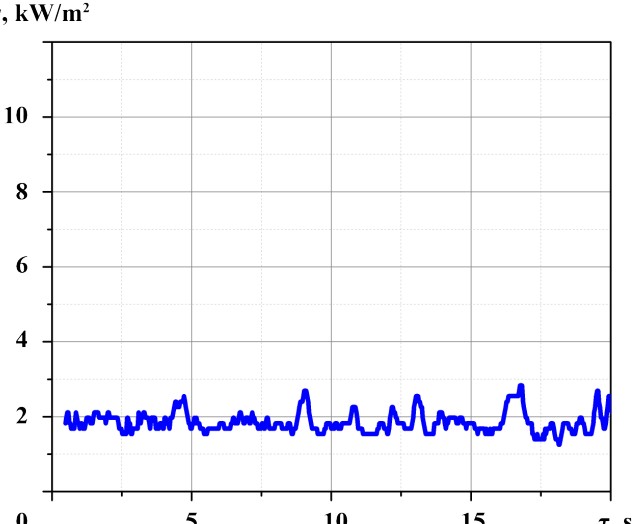

**Figure 5.** Time-dependent heat flux graph by GHFS for the regime without cooling (at the $\dot{G}_w = 0$).

*2.3. Experimental Setup*

As a result of the experiments presented in Section 2.2, the requirements for the experimental setup were:

- The pipe material must be exchanged for a less thermal conductivity one. The copper's high thermal conductivity led to temperature equalization over the pipe surface and increased the model thermal inertia which reduced the information content of the temperature measurement;
- The measuring section length must be increased to hydrodynamic flow stabilization;
- The GHFS cross area should be increased to promote the generated thermoEMF, but it is necessary to minimize GHFS width. The purpose of the study is to determine the heat flux distribution over the horizontal pipe surface, and it is necessary to reduce the azimuthal angle $\varphi$ at which the heat flux is averaged;
- To eliminate distortions in the natural condensate flow, it is necessary to minimize the number of GHFS and thermocouples and develop a method for removing wires from the heat exchange surface.

An experimental model for studying the heat flux distribution during the saturated steam condensation on the outer surface of a horizontal pipe was made according to the "pipe in pipe" scheme (Figure 6). The inner tube was made of stainless steel with a thermal conductivity of about 15 W/(m×K) and an outer diameter of about 20 mm. The case was made of reinforced rubber pipe with an inner diameter of 60 mm. The stainless steel pipe was fixed in the case by rubber plugs. In the experiments, saturated steam at the atmospheric pressure was fed into the annular gap between the tubes; its mass flow rate was 2.8 g/s. Cooling water with a temperature of 293 K and a flow rate of 200 mL/s entered the stainless steel tube. The condensate was removed to a condensate collector where its flow rate was determined from the mass of condensate. The length of the measurement section was about 800 mm.

Measurements were made with the single-bismuth GHFS (Table 3) and L-type thermo-couples mounted on the same stainless steel pipe generatrix at a distance of 400 mm from the flanges. The GHFS was installed in a milled recess flush with the outer pipe surface (Figure 7) to reduce distortion during the condensate flow. The width of the GHFS was about 2.5 mm which led to the averaging of its readings over 9° of the azimuth angles $\varphi$ of the pipe.

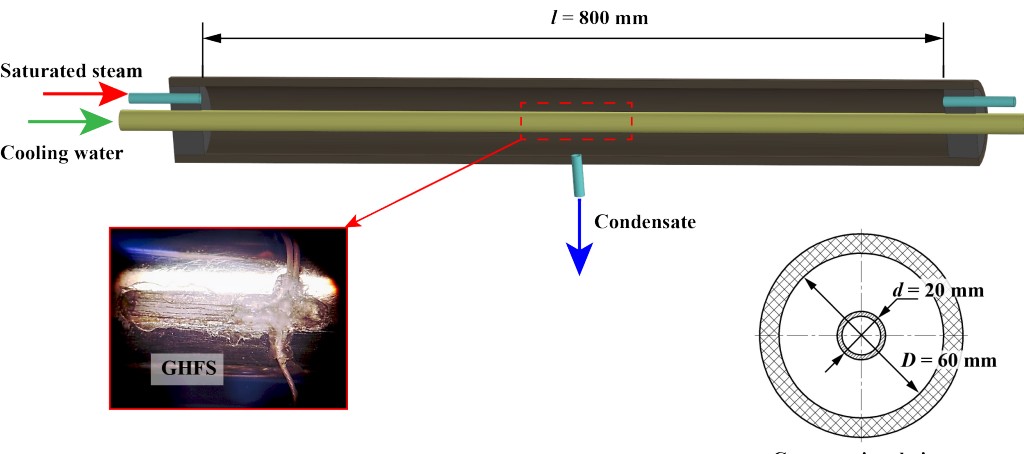

**Figure 6.** The experimental model.

**Table 3.** GHFS specifications.

| Sensor Type | Material | Sizes, mm | Volt–Watt Sensitivity $S_0$, mV/W |
|:---:|:---:|:---:|:---:|
| GHFS | single-crystal bismuth | $2.5 \times 10 \times 0.2$ | 2.65 |

To minimize disturbances of the natural condensate flow, the wiring from the sensor and the thermocouples were routed along the guide strings (Figure 7) located 7 mm from the pipe surface. All instrument and thermocouple wiring were brought out via the rubber plug.

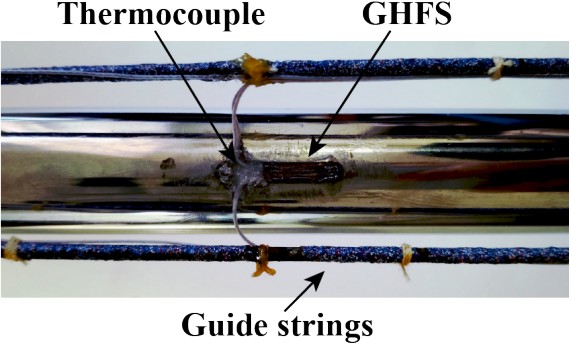

**Figure 7.** Measuring section elements.

The experiments were carried out with above-described test section located horizontally. The pipe rotation about the axis in the range of azimuthal angle of $\varphi = 0\ldots180°$ with a step of $15°$ made it possible to estimate the distribution of the local heat flux area along the pipe's circumference. The azimuthal angle reference began from the top pipe generatrix. This made it possible to estimate the distribution of the heat flux and the pipe's surface temperature along the horizontal pipe circumference by using the installation of one pair of the GHFS and the thermocouple.

*2.4. Uncertainty Analysis*

The uncertainty was calculated according to ISO/IEC GUIDE 98-1:2009—Uncertainty of Measurement [33]. The standard uncertainty of the heat flux per unit area was calculated as:

$$u_q = \sqrt{\left(\frac{\partial q}{\partial E} u_E\right)^2 + \left(\frac{\partial q}{\partial S_0} u_{S_0}\right)^2 + \left(\frac{\partial q}{\partial A} u_A\right)^2} =$$

$$= \sqrt{\left(\frac{1}{S_0 A} u_E\right)^2 + \left(-\frac{E}{S_0^2 A} u_{S_0}\right)^2 + \left(-\frac{E}{S_0 A^2} u_A\right)^2} = 3670 \ \text{W/m}^2, \tag{7}$$

where $u_E$ is the thermo-EMF measurement uncertainty, $u_{S_0}$ is the standard uncertainty of the GHFS volt–watt sensitivity, and $u_A$ is the standard uncertainty of the GHFS area. The uncertainty values of the input estimates are given in Table 4.

**Table 4.** Uncertainty budget for heat flux per unit area.

| Quantity, $X_i$ | Estimate, $x_i$ | Standard Uncertainty, $u(x_i)$ | Uncertainty Contribution, $u(y_i)$ |
|:---:|:---:|:---:|:---:|
| $E$ | 9341 μV | 35.0 μV | 528 W/m$^2$ |
| $S_0$ | 2650 μV/W | 67.4 μV/W | 3582 W/m$^2$ |
| $A$ | $25 \times 10^{-6}$ m$^2$ | $10.7 \times 10^{-8}$ m$^2$ | 605 W/m$^2$ |

The expanded uncertainty of the heat flux per unit area is obtained by multiplying the standard uncertainty by a coverage factor $k = 2$.

$$U = k u_q = 7340 \ \text{W/m}^2. \tag{8}$$

The heat flux per unit area relative uncertainty is about 5.2%.

The standard uncertainty of the HTC is calculated as:

$$u_h = \sqrt{\left(\frac{\partial h}{\partial q} u_q\right)^2 + \left(\frac{\partial h}{\partial \Delta T} u_{\Delta T}\right)^2} = \sqrt{\left(\frac{1}{\Delta T} u_q\right)^2 + \left(-\frac{q}{\Delta T^2} u_{\Delta T}\right)^2} = 258 \ \text{W/(m}^2 \ \text{K)}. \tag{9}$$

The uncertainty values of the input estimates are given in Table 5.

**Table 5.** Uncertainty budget for HTC.

| Quantity, $X_i$ | Estimate, $x_i$ | Standard Uncertainty, $u(x_i)$ | Uncertainty Contribution, $u(y_i)$ |
|:---:|:---:|:---:|:---:|
| $q$ | 141,060 W/m$^2$ | 3670 W/m$^2$ | 142.8 W/(m$^2$ K) |
| $\Delta T$ | 25.6 K | 1 K | 215.2 W/(m$^2$ K) |

The expanded uncertainty of the HTC is about

$$U = k u_h = 516 \ \text{W/(m}^2 \ \text{K)}. \tag{10}$$

The HTC relative uncertainty is about 9.4%.

According to the results of calculating the uncertainty of measuring the heat flux and the HTC, it can be concluded that the proposed method provides acceptable accuracy in the study of heat transfer during condensation. Measurement uncertainty is comparable to other methods described in the Introduction, but the benefits of direct heat flux measurement open up possibilities for researchers.

## 3. Results

We consider the distribution of heat transfer parameters over the pipe' surface. The heat flux and the temperature of the outer surface of the stainless steel pipe were measured simultaneously. The average values are summarized in Table 6. As shown above, the measurement step was 15°.

**Table 6.** The time-averaged local heat flux and surface temperature along the pipe circumference.

| $\varphi$, ° | $q$, kW/m$^2$ | $T$, K |
|---|---|---|
| 0 | 117.57 | 350.0 |
| 15 | 116.32 | 350.5 |
| 30 | 131.13 | 349.6 |
| 45 | 176.41 | 348.9 |
| 60 | 181.13 | 346.9 |
| 75 | 173.34 | 348.2 |
| 90 | 156.86 | 346.9 |
| 105 | 161.15 | 346.9 |
| 120 | 157.94 | 347.0 |
| 135 | 137.93 | 346.9 |
| 150 | 159.04 | 345.5 |
| 165 | 95.73 | 344.1 |
| 180 | 94.23 | 344.1 |

The analysis of the time-averaged heat flux distribution indicates the enhancement of steam condensation heat transfer in the range of $\varphi = 45\ldots75°$. The GHFS recorded the formation of a bottom zone in the range of $\varphi = 150\ldots180°$.

The similar analysis of the time-averaged temperature distribution indicates the presence maximum at the upper region of the pipe in the range of $\varphi = 0\ldots15°$ and a monotonic temperature decrease with an increase in the azimuth angle $\varphi$. It is impossible to unambiguously determine the areas of heat transfer enhancement or the organization of a stagnant zone by temperature measurement.

The time-dependent heat flux graphs at the different azimuthal angle are shown in Figure 8. All plots show fluctuations in the heat flux which confirm the non-stationarity of heat transfer during condensation on the horizontal pipe.

On the upper generatrix of the pipe at $\varphi = 0°$ (Figure 8a), as in the experiments described in Section 2.2, there is no wave formation and the effect of steam velocity on the movement of condensate. It turns out that the heat flux fluctuations are associated with the formation and run-off of the condensate from the pipe surface.

The highest average heat flux of 181.13 kW/m$^2$ corresponds to the pipe rotation on azimuthal angle of $\varphi = 60°$. The time heat graph (Figure 8c) shows periodic fluctuations of heat flux in which the amplitude does not exceed 20% of the average value. Therefore, there is an assumption that the fluctuations are due to the condensate flowing in the form of separate rivulets in the upper region.

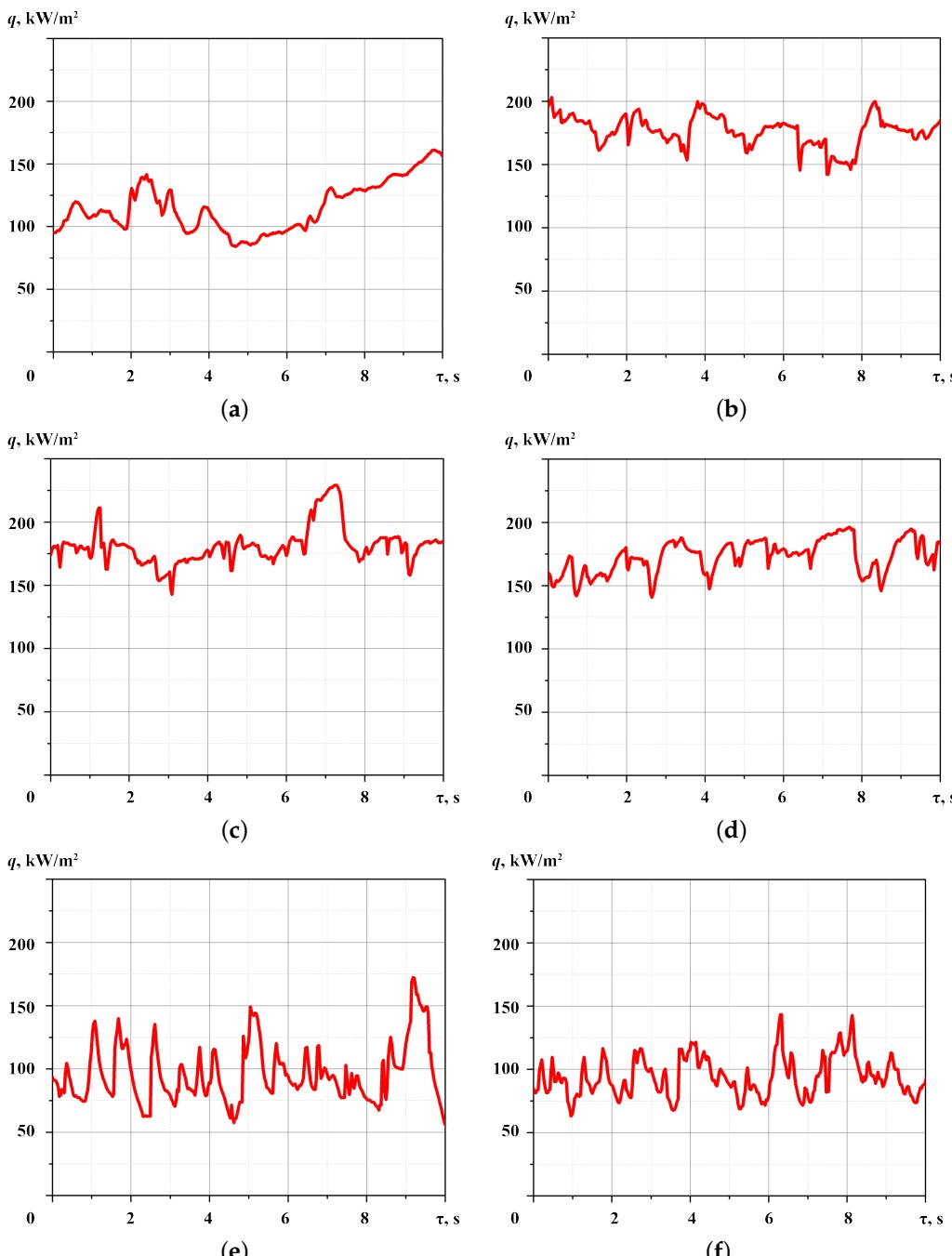

**Figure 8.** Time-dependent heat flux graph for saturated steam condensation on the horizontal pipe at the azimuthal angle $\varphi$ of: (**a**)—0°; (**b**)—45°; (**c**)—60°; (**d**)—75°; (**e**)—165°; (**f**)—180°.

The time-averaged heat flux decreases to 94.23 kW/m² when the measuring section is rotated by $\varphi$ = 180°. The amplitude of heat flux fluctuations is up to 35% of the average value. There is an assumption that this effect is associated with the condensate accumulation in the lower region of the pipe and its run-off from the pipe surface in the form of separate conglomerations.

Critical remarks to the obtained results are revealed.

1.  The experimental conditions do not provide for the organization of film condensation on the surface of a horizontal pipe. According to the fluctuations on the heat flux graphs (Figure 8), the condensate flows down from the pipe surface in the form of separate rivulets. It is necessary to add visual observation of the condensate flow for an explanation of fluctuations in the heat flux recorded using GHFS.

2. There is no generally accepted theoretical model for calculating the heat flux during not-filmwise and not-dropwise condensation. Comparison of the results with the Nusselt model for film condensation is irrelevant. Therefore, in continued investigation, the regime conditions should be expanded to achieve film condensation on a horizontal pipe.

The main advantage of the proposed approach lies in the versatility of GHFS, the application of which is limited only by the melting point of bismuth. The gradient heatmetry can be used to study the heat flux during condensation but must be supported by temperature measurements and visual observation of the condensate flow to unambiguously evaluate the results.

## 4. Discussion

The generalized experimental results are presented in the polar coordinates on the angular heat flux graph—local heat flux distribution over the horizontal pipe (Figure 9).

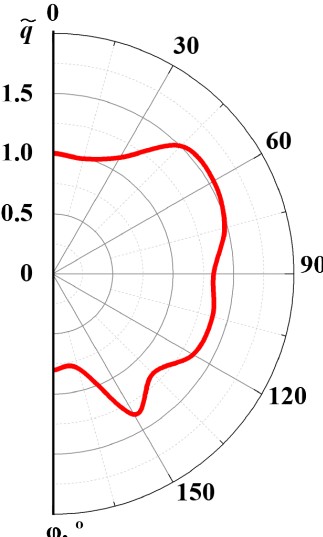

**Figure 9.** Angular heat flux graph for saturated steam condensation on the outer surface of a horizontal pipe.

The heat flux is transformed into a relative form: the local heat flux is related to its value on the top generatrix of the pipe:

$$\widetilde{q} = \frac{q(\varphi)}{q(0)}, \tag{11}$$

where $q(\varphi)$ is the local heat flux measured on the azimuth angle $\varphi$, and $q(0)$ is its value on the top generatrix of the pipe ($\varphi = 0°$).

The results indicate the stagnant zone development in the range of $\varphi = 150...180°$, which is explained by the condensate accumulation and the sub-bottom area development in the lower region of the horizontal pipe.

A similar processing of the results was applied to temperature measurements. Figure 10 shows the relative temperature distribution during the saturated steam condensation over the horizontal pipe. The local surface temperature is reduced to the relative form:

$$\widetilde{T} = \frac{T(\varphi)}{T(0)}, \tag{12}$$

where $T(\varphi)$ is the local pipe temperature on the azimuth angle $\varphi$, and $T(0)$ is its value on the pipe top generatrix ($\varphi = 0°$).

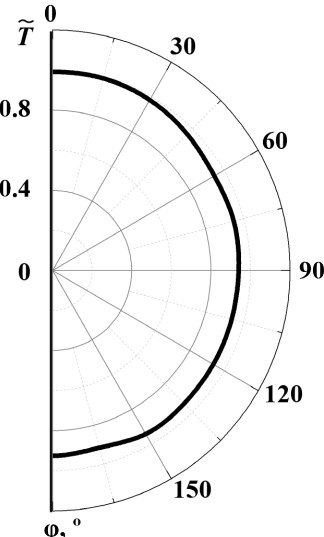

**Figure 10.** Relative temperature distribution over the horizontal pipe surface.

Comparing Figures 9 and 10 confirms the gradient heatmetry informativeness. The temperature measurement does not allow us to determine the condensate accumulation region. At the same time, the gradient heatmetry results allow us to propose ways to enhance heat transfer. For example, it is possible to use specially shaped fins only in the sub-bottom area.

Simultaneous measurement of heat flux and temperature made it possible to determine the HTC distribution over the horizontal pipe surface. Figure 11 shows relative HTC distribution for a horizontal pipe obtained experimentally.

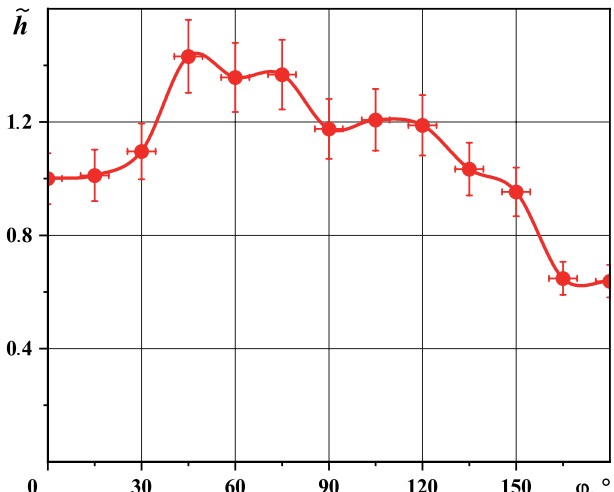

**Figure 11.** The relative HTC distribution over the horizontal pipe surface.

Comparison of experimental and calculation results is ambiguous:

1. The experiment result indicates that the HTC maximum corresponds to the region of $\varphi = 30...75°$.
2. The HTC becomes less than the value on the upper generatrix only in the lower pipe region in the range of $\varphi = 150...180°$.
3. Theoretically, the HTC is equal to zero on the lower pipe generatrix. However, in practice this is not possible because then the thickness of the condensate film is equal to an infinitely large value. The experiment results indicate HTCs decrease in the lower pipe region which is associated with a sub-bottom zone formation.

We assume that under the experimental conditions condensate film was not formed, and the condensate flowed down from the horizontal pipe surface in separate rivulets. This assumption is based on the behavior of time–heat graphs and the heat flux fluctuations.

## 5. Conclusions

In the presented paper, a fundamentally new approach to determining the heat flux during condensation is applied. The heat flux per unit area is measured "directly" using a GHFS.

The studies were carried out in the non-film condensation regime, which confirms the fluctuations on the heat flux graphs. We assume the condensate run-off from the horizontal pipe is in the form of separate rivulets. The results indicate a significant non-stationarity of heat transfer during the saturated steam condensation on the outer surface of a horizontal pipe. The heat flux fluctuations can be explained by the condensate formation, not film flow and run-off from the horizontal pipe surface. The average surface heat flux during condensation on a horizontal pipe was about 141.06 kW/m$^2$. The local heat flux along the pipe circumference varied in a range from 94.23 to 181.13 kW/m$^2$. The increase and decrease in the heat flux was associated with a change in the condensate flow. Thus, a decrease in the heat flux occurred in the lower part of the pipe, where condensate accumulates. The proposed approach provides a relative expended uncertainty of local heat flux per unit area measurement of about 5.2% and in the calculation of the HTC of about 9.4%.

Comparison of the results of gradient heatmetry and temperature measurements reveals the advantage of GHFS. Temperature measurement with L-type thermocouples does not allow evaluation of condensate distribution and identification of liquid accumulation areas.

The main advantage of gradient heatmetry is the ability to choose the type and size of the sensor in accordance with the experiment conditions. A special series of experiments was carried out to determine the optimal type and sizes of GHFS for experiments. It was found that GHFS from single-crystal bismuth is suitable for studying heat transfer during condensation. The high sensitivity of the GHFS (about 2.56 mV/W) makes it possible to reduce its cross area to 25 mm$^2$.

Experiments have shown that future studies should be carried out by combining the gradient heatmetry, temperature measurement and optical methods. For an unambiguous conclusion, a single experiment of the thickness of the condensate film with the heat flux per unit area and HTC during condensation on the outer surface of horizontal pipe is needed. The gradient heatmetry will find application both in thermophysical experiments and in applied problems related to the creation and heat exchanger improvement for various purposes.

The proposed innovative approach will make it possible to measure heat flux under various condensate flow regimes, different surface configurations, and condensation of various working fluids. The experimental study of heat transfer during condensation using gradient heatmetry is important for industry. The expansion of the experiment parameters will improve the engineering methods for calculating heat exchangers and condensers. The application of the method in industry will make it possible to create a unique system for monitoring the state of condensation surfaces.

**Author Contributions:** Conceptualization, E.R.Z. and V.Y.M.; methodology, E.R.Z. and V.Y.M.; validation, E.R.Z.; formal analysis, V.Y.M.; investigation, E.R.Z.; resources, E.R.Z. and V.Y.M.; data curation, E.R.Z.; writing—original draft preparation, E.R.Z.; writing—review and editing, E.R.Z.; visualization, E.R.Z.; supervision, V.Yu.; project administration, V.Y.M.; funding acquisition, E.R.Z. and V.Y.M. All authors have read and agreed to the published version of the manuscript.

**Funding:** This research was funded by Russian Science Foundation grant number 22-29-00152, https://rscf.ru/project/22-29-00152/ (accessed on 13 November 2022).

**Informed Consent Statement:** Not applicable.

**Data Availability Statement:** Not applicable.

**Conflicts of Interest:** The authors declare no conflict of interest.

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
