# Peer review of "Heat Transfer Investigation during Condensation on the Horizontal Pipe"

_inventions, doi:10.3390/inventions8010002_

Round 1

Reviewer 1 Report

The topic is interesting to the relevant research field, however, the novel feature of the work is not clear showed. From this perspective, it needs minor revision. The authors have conducted a brief work but some important things should be included in the revised manuscript. Only then the work can be accepted

The abstract is too general; the abstract should contain significant results with a certain value or percentage.

Figures need to be improved ,they are not visible clearly.

It is suggested to carry out extensive literature survey.

At the end of each section in the results, authors should present concluding remarks with critical discussion in their own words..

Conclusion should be re-written

Author Response

Thank you for your valuable comments and attention to the manuscript.

Point 1: The abstract is too general; the abstract should contain significant results with a certain value or percentage.

Response 1: According to your remark, the abstract indicates the average surface heat flux of condensation on a horizontal pipe; the range of azimuthal angles at which condensate accumulation was observed and the percentage of heat flux reduction in the lower part of the pipe. Also, now the abstract includes the values of the measurement uncertainty of the heat flux and heat transfer coefficients during condensation.

Point 2: Figures need to be improved; they are not visible clearly.

Response 2: Improved the quality of pictures included in the manuscript.

Point 3: It is suggested to carry out extensive literature survey.

Response 3: The literature review has been expanded. Various experimental approaches to the study of heat transfer during condensation are considered.

Point 4: At the end of each section in the results, authors should present concluding remarks with critical discussion in their own words.

Response 4: The results section contains concluding remarks with a critical discussion of the proposed approach.

Critical remarks to the obtained results are revealed.

  • The experimental conditions do not provide for the organization of film condensation on the surface of a horizontal pipe. According to the fluctuations on the heat graphs, the condensate flows down from the pipe surface in the form of separate rivulets. It is necessary to add visual observation of the condensate flow for an explanation of fluctuations in the heat flux recorded using GHFS.
  • There is no generally accepted theoretical model for calculating the heat flux during not-filmwise and not-dropwise condensation. Comparison of the results with the Nusselt model for film condensation is irrelevant. Therefore, in the investigation continuation, the regime conditions should be expanded to achieve film condensation on a horizontal pipe.

Point 5: Conclusion should be re-written

Response 5: The conclusion has been reformulated.

With respect and gratitude, the authors

Reviewer 2 Report

In the manuscript, an experimental investigation of condensation heat transfer by gradient heatmetry is presented. The experimental work is always interesting and requires much work for proper experiment design, calculation and interpretation of results. In this case, it is unfortunately not the case.

There are several weaknesses: 

- The measurement procedure is not really measuring condensation on a tube but something similar on the "heat meter" which may be far away from the proper tube condensing heat transfer result. There is a heat resistance of this device which is not really calculated. Besides the device's (GHFS)  presence causes breaks in the condensing film which strongly influences the results. 

- To properly assess the heat and temperature distribution in this case a full 3D ANSYS/FLUENT analysis is necessary. This may be also helpful for the third significant weakness

- The third very significant weakness is the lack of uncertainty analysis. This analysis for this case requires not only all devices' accuracy consideration but the uncertainties due to the heat resistance of the ( GHFS) elements. 

The comparison with Nusselt is in this case totally irrelevant since Nusselt theory addresses pure tubes without any "disturbances" on the surface.

Author Response

Thank you for your valuable comments and attention to the manuscript.

Point 1: The measurement procedure is not really measuring condensation on a tube but something similar on the "heat meter" which may be far away from the proper tube condensing heat transfer result. There is a heat resistance of this device which is not really calculated. Besides the device's (GHFS) presence causes breaks in the condensing film which strongly influences the results.

Response 1: The gradient heat flux sensor (GHFS) was used as a «heat meter» in our investigation. The heat flux during condensation was estimated from the GHFS signal. The additional file presents the calculation of the thermal resistance of the pipe surface and with a sensor installed on the pipe surface. The difference in thermal resistances leads to a difference between the temperature of the outer wall of the pipe without a sensor and the temperature on the surface of the GHFS of about 2.3%. The relative standard uncertainty of temperature measurement given in the uncertainty budget is about 3.9%. The difference in thermal resistance of the pipe and the «GHFS-pipe» system will not lead to large errors in the measurement of the heat flux during condensation.

Point 2: To properly assess the heat and temperature distribution in this case a full 3D ANSYS/FLUENT analysis is necessary. This may be also helpful for the third significant weakness

Response 2:  Thank you very much for your advice. In future studies on the fluctuation’s frequency of the heat flux, thermal resistance will be evaluated in the 3D ANSYS/FLUENT. The thermal resistances are calculated according to generally accepted formulas for the stationary heat conduction.

Point 3: The third very significant weakness is the lack of uncertainty analysis. This analysis for this case requires not only all devices' accuracy consideration but the uncertainties due to the heat resistance of the (GHFS) elements.

Response 3: According to your comment, an uncertainty analysis section is included in the manuscript.

 Point 4: The comparison with Nusselt is in this case totally irrelevant since Nusselt theory addresses pure tubes without any "disturbances" on the surface.

Response 4: Thanks for the valuable advice. Comparison of the results with the Nusselt model is completely excluded from the manuscript as incorrect.

With respect and gratitude, the authors.

Round 2

Reviewer 1 Report

Still introduction is lacking so kindly add few papers in introduction section 

Conclusion still improvement is required

Author Response

Thanks for reviewing the manuscript.

Point 1: Still introduction is lacking so kindly add few papers in introduction section.

Response 1: The number of articles included in the literature review has been increased.

Point 2: Conclusion still improvement is required.

Response 2: The conclusion is expanded.

Reviewer 2 Report

My first concern is not answered correctly. Only the Authors state that the "system will not lead to large errors in the measurement of the heat flux during condensation."  This is not good enough for validation. However, the idea of a heat meter and experimental analysis with clarified uncertainty is interesting and worth publishing. My other concerns were addressed. 

Author Response

Point 1: My first concern is not answered correctly. Only the Authors state that the "system will not lead to large errors in the measurement of the heat flux during condensation."  This is not good enough for validation. However, the idea of a heat meter and experimental analysis with clarified uncertainty is interesting and worth publishing. My other concerns were addressed.

Response 1: We apologize for the incompleteness of the answer to your question. To solve this problem, the experimental setup will be improved in future investigation. To do this, thermocouples will be installed near the gradient heat flux sensor on the inner and outer surfaces of the pipe opposite each other. This will definitely allow us to calculate the heat flux and compare it with the one measured using gradient heatmetry.

Thank you for your valuable advice. We believe that solving this problem will improve the accuracy of our investigations.